# Molecular Characteristics of RAGE and Advances in Small-Molecule Inhibitors

**DOI:** 10.3390/ijms22136904

**Published:** 2021-06-27

**Authors:** Hyeon Jin Kim, Mi Suk Jeong, Se Bok Jang

**Affiliations:** 1Department of Molecular Biology, College of Natural Sciences, Pusan National University, Jangjeon-dong, Geumjeong-gu, Busan 46241, Korea; khjkhj0903@naver.com; 2Insitute for Plastic Information and Energy Materials and Sustainable Utilization of Photovoltaic Energy Research Center, Pusan National University, Jangjeon-dong, Geumjeong-gu, Busan 46241, Korea

**Keywords:** RAGE, multi-ligands, disease, drug, inhibitor

## Abstract

Receptor for advanced glycation end-products (RAGE) is a member of the immunoglobulin superfamily. RAGE binds and mediates cellular responses to a range of DAMPs (damage-associated molecular pattern molecules), such as AGEs, HMGB1, and S100/calgranulins, and as an innate immune sensor, can recognize microbial PAMPs (pathogen-associated molecular pattern molecules), including bacterial LPS, bacterial DNA, and viral and parasitic proteins. RAGE and its ligands stimulate the activations of diverse pathways, such as p38MAPK, ERK1/2, Cdc42/Rac, and JNK, and trigger cascades of diverse signaling events that are involved in a wide spectrum of diseases, including diabetes mellitus, inflammatory, vascular and neurodegenerative diseases, atherothrombosis, and cancer. Thus, the targeted inhibition of RAGE or its ligands is considered an important strategy for the treatment of cancer and chronic inflammatory diseases.

## 1. Introduction

RAGE (receptor for advanced glycation end-products) was first isolated from the human lung library in 1992 and noted for its ability to act as a receptor for advanced glycation end products (AGEs). RAGE is a transmembrane protein of the immunoglobulin (Ig) superfamily of cell surface molecules [1], and interacts with multiple ligands that mediate cellular responses to a range of DAMPs (damage-associated molecular pattern molecules), such as AGEs, the S100 group of proteins, HMGB1 (high mobility group box-1 protein), amyloid β, and DNAs, and also acts as an innate immune sensor of PAMPs (pathogen-associated molecular pattern molecules), such as bacterial LPS, respiratory viruses, viral and parasitic proteins, and bacterial DNA [2,3,4,5,6,7,8,9,10,11,12]. Ligand stimulation of RAGE activates signal transduction pathways, such as the diaphanous-related formin 1 (DIAPH1), mitogen-activated protein kinase (MAPK), phosphatidylinositol 3-kinase (PI3K)/AKT, and Toll-interleukin 1 receptor domain-containing adaptor protein (TIRAP) pathways, which result in RAGE-dependent NF-κB activation [13,14,15,16,17,18,19]. RAGE is expressed in many cell types, including endothelial, vascular smooth muscle, and cancer cells monocyte/macrophages, granulocytes, and adipocytes [20]. Upregulated RAGE expression has been reported in diabetes mellitus, atherosclerosis, rheumatoid arthritis, Alzheimer’s disease (AD), cardiovascular diseases (CVDs), and immune/inflammatory diseases [21,22,23,24,25], and has also been shown to be related to the developments and progressions of different cancer types [26].

## 2. Structure and Isoforms of RAGE

RAGE is a 50–55 kDa glycosylated protein that contains an extracellular (amino acids 23–342), a hydrophobic transmembrane (residues 343–363), and a cytoplasmic (residues 363–404) domain. The extracellular structure of RAGE is composed of a variable (V) immunoglobulin (Ig) domain (residues 23–116) and two constant C1 (residues 124–221) connected to C2 (residues 227–317) Ig domains by a flexible seven amino acid linker [27]. Its V domain consists of eight strands (A’, B, C, C’, E, F, and G) connected by six loops that form two beta-sheets linked by a disulfide bridge between Cys38 (strand B) and Cys99 (strand F) [28]. The molecular surfaces of V-C1 domains are covered by a hydrophobic cavity and contain many highly positively charged Arg and Lys residues [17]. In contrast, the C2 domain is composed of acidic amino acids and carries a negative surface charge (Figure 1) [29]. Multiple RAGE ligands contain highly negatively charged regions and can bind to the positively charged V-C1 domain [7]. Ding Xu reported that heparan sulfate plays a crucial role in stabilizing RAGE homodimerization the self-association of V-V domains and RAGE hexamerization [30]. In its monomeric state, RAGE has only weak affinity for several ligands, and thus, it appears that its multimerization is necessary for ligand binding. RAGE oligomerizations through its C1-C1 domains, C2-C2 domains, and/or TM helix dimerization are important steps for RAGE signaling after ligand binding [30,31,32]. The transmembrane helical structure of RAGE contains the meticulously conserved GxxxG motif, which promotes helix-helix homodimerization and may be involved in signal transduction [33]. The cytoplasmic domain of RAGE exhibits high sequence identity with primates and rodents, which is essential for RAGE ligand-mediated signal transduction [1], and its cytoplasmic domain contains a highly acidic region that is capable of binding several molecules. In fact, truncation of this domain abolishes downstream RAGE signaling and attenuates RAGE-associated pathologic effects [34,35].

The human RAGE gene is located on chromosome 6 in the MHC (major histocompatibility complex) class III region, which contains many genes that impact the adaptive and innate immune systems [29]. More than 20 RAGE isoforms with diverse biological functions have been found to result from alternative splicing [36]. In addition, polymorphisms in RAGE have been suggested to be potential biomarkers in RAGE-relevant diseases [37,38], and the RAGE transcript has been identified as the target for the productions of alternative splicing generates isoforms, such as full-length RAGE (FL-RAGE), dominant-negative RAGE (DN-RAGE, residues 23–363), N-truncated RAGE (N-RAGE, 124–404), C-truncated soluble RAGE (sRAGE, 23–342), and splice variant endogenous secretory RAGE (esRAGE) [39]. The functions of these truncated RAGE isoforms have yet to be elucidated, but it has been established the dysregulations of RAGE isoforms and their ligands lead to the development of a number of human diseases [40,41]. Soluble RAGE may competitively inhibit RAGE-ligand-mediated signaling, and a low sRAGE level has been suggested to be a biomarker for diseases [7]. On the other hand, serum levels of sRAGE in diabetes, sepsis, and end-stage renal disease (ESRD) are elevated [42,43].

## 3. RAGE as a Multi-Ligand Receptor

RAGE binds diverse classes of ligands, such as HMGB1, S100 calcium-binding protein/calgranulin, amyloid-β, and lysophosphatidic acid (LPA) [3,5,22,44,45,46,47]. RAGE expression can also increase DNA internalization and augment the Toll-like receptors (TLR) response through TLR9 [48,49]. Ligand engagement of RAGE activates multiple signaling pathways, including those of ERK, AKT, STAT3, JNK and MAPK, which result in the activations of transcription factors, including NF-κB [50,51]. Furthermore, interactions between RAGE and multiple ligands upregulate RAGE through positive feedback loops [52], and following RAGE activation are expressed on various cell types, including endothelial cells, vascular smooth muscle cells, lymphocytes, neurons, monocytes/macrophages, and podocytes [20,35,53,54,55]. Moreover, ligand-RAGE interactions are involved in the pathogeneses of diabetes mellitus, chronic renal failure, rheumatoid arthritis, atherosclerosis, neurodegenerative diseases, cancer, immune/inflammatory responses, and aging [56,57,58,59,60,61,62,63]. Ligands of RAGE are listed in Table 1.

### 3.1. Endogenous RAGE

Many types of AGE have been identified, and aging leads to accumulations of AGEs in tissues and plasma [71]. *N_ɛ_*-carboxymethyl-lysine (CML) and *N_ɛ_*-carboxyethyl-lysine (CEL) are found in human tissue and blood plasma and bind to the V domain of RAGE. Interactions between AGEs and RAGE induce the expressions of pro-inflammatory cytokines and chemokines, such as TNF-α, IL-1β, and CCL2 [72,73], and have been linked with the complications of diabetes, chronic inflammation, Alzheimer’s disease, and cancer [28]. Methylglyoxal (2-oxoaldehyde) is a precursor of AGEs and a reactive α-oxaldehyde. RAGE binds to three structural isomers of methylglyoxal-derived hydroimidazolones (MG-H), that is, MG-H1 [Nδ -(5-hydro-5-methyl-4-imidazolon-2-yl) ornithine], MG-H2 [5-(2-amino-5-hydro-5-methyl-4-imidazolon-1-yl) norvaline], and MG-H3 [5-(2-amino-4-hydro-4-methyl-5-imidazolon-1-yl)- norvaline], and binding between the V-domain of RAGE and MG-H increases the phosphorylation of c-Jun *N*-terminal kinase (JNK) in vitro [74].

The S100 protein family contains 25 members with different expression patterns, functions, and oligomeric states, and S100B, S100A1, S100A2, S100A4, S100A5, S100A5, S100A6, S100A7, S100A8/A9, S100A11, S100A12, and S100P have been shown to interact with RAGE in vivo [2]. The S100 proteins are small proteins (9–13 kDa) that bind calcium via EF-hand domains and act as calcium sensors, which participate in calcium signal transduction. They are also involved in the regulation of several cellular processes, such as cell differentiation and progression in invertebrates. Most S100 calcium-binding protein genes are located in human chromosome 1q21, which tends to exhibit physical chromosomal rearrangements [75]. S100 proteins interact and regulate various proteins involved in the dynamics of cytoskeletal constituents, calcium homeostasis, cell growth, and differentiation. S100B is mainly expressed in the brain and is well expressed and secreted by astrocytes, Schwann cells, and oligodendrocytes [64,76,77]. Extracellular S100B proteins bind to the RAGE V-domain and recruit PI3K/AKT and NF-κB [78], and these interactions induce trophic and inflammatory responses by neurons and carcinogenesis [78,79,80]. S100A1 is primarily expressed in the heart and only marginally expressed in other tissues [81]. S100B and S100A1 were reported to interact with RAGE on cell surfaces, inducing neurite outgrowth, and increase cell survival in a HMGB1 dependent manner [82]. S100A2 has been shown to interact with and increase the transcriptional activity of tumor suppressor protein p53, and is downregulated in many cancers, including prostate, oral, melanoma, lung, and breast cancer [83,84,85,86,87,88]. However, it is upregulated in other cancers, such as gastric, esophageal squamous carcinoma, non-small lung carcinoma, and ovarian cancer [89,90,91,92]. Leclerc demonstrated that S100A2 interacts with the V-domain of RAGE [2]. The binding between S100A4 and RAGE increased the production of matrix metalloproteinase 13 (MMP-13), which requires modulation of intracellular calcium levels [93]. S100A5 interacts with the RAGE V-domain in a calcium-dependent manner and is highly expressed in astrocytic tumors [94]. S100A6 is found at high levels in the lungs, kidneys, muscles, spleen, and the brain, and is overexpressed and plays important roles in many cancers including melanoma, lung cancer, hepatocellular carcinoma, colorectal cancer, and gastric cancer [84,95,96,97,98,99,100,101]. S100A6 interacts with the V and C2 domains of RAGE and activates JNK signaling [78]. RAGE-S100A7 (psoriasin) mediates chemotaxis and is involved in the regulation of pro-inflammatory and antimicrobial functions [102]. S100A8 (calgranulin A, also known as MRP8) and S100A9 (calgranulin B, also known as MRP14) are strongly expressed by cells of myeloid origin, epithelial cells, and keratinocytes during inflammation. S100A8/A9 can form heterodimers or hetero-tetramers in the absence or presence of calcium, respectively [103,104,105,106]. High levels of S100A8/9 have been found in Alzheimer’s disease, rheumatoid arthritis, Crohn’s disease, cystic fibrosis, and several cancers, such as colorectal carcinoma, prostate cancer, and gastric cancer [107,108,109,110,111,112,113]. Although the RAGE/S100A8/A9 interaction is unclear, it is known to promote cell growth via p38MAPK, p44/42 kinase, and NF-κB in tumor cells [114,115]. S100A11 levels are elevated in prostate, breast, and pancreatic cancer; however, S100A11 acts as a tumor suppressor in bladder and renal carcinomas. Furthermore, S100A11-RAGE signaling modulates the pathogenesis of osteoarthritis (OA) by regulating differentiation to hypertrophy [116,117,118,119,120,121]. S100A12 (calgranulin A) is highly expressed in inflammatory diseases, such as cystic fibrosis, atherosclerosis, psoriasis, rheumatoid arthritis, Crohn’s disease, and Kawasaki disease [122,123,124,125,126,127]. S100A12 promotes intracellular signal transduction in neurons involving the MAPK and phospholipase C pathways [128]. SPR showed S100A12 interacts with the V-domain of RAGE [2]. The interaction between S100A13 and RAGE has not been revealed, though it is known S100A13 translocates from nucleus to cytoplasm and that this leads to the extracellular secretion of endothelial cells in a RAGE dependent-manner [129]. S100A13 mRNA has been reported in many tissues and organs including kidneys, ovaries, spleen, brain, and heart. This AGE is a marker of angiogenesis in human astrocytic gliomas and invasive lung cancer [130,131,132,133]. S100P is present in breast, gastric, ovarian, pancreatic, and prostate cancer and colorectal carcinoma [134,135,136], and the S100P-RAGE interaction activates ERK and NF-κB signaling pathways in NIH3T3 cells [137].

HMGB1 is a highly conserved nuclear protein that acts as a chromatin-binding factor, and this binding leads to the bending of DNA and the promotion of DNA transcription [138]. HMGB1 is passively released by necrotic cells and actively secreted by inflammatory cells. Extra HMGB1 binds with high affinity to several receptors including TLR-2, TLR-4, and RAGE on endothelial, smooth muscle and cancer cells and neutrophils [65,139,140]. HMGB1 overexpression is a hallmark of sepsis, arthritis, neurodegeneration, aging, angiogenesis, and cancer development and metastasis [141,142,143,144,145]. Interaction between RAGE and HMGB1 promotes the activations of tumor cell signaling pathways, such as those of ERK1/2, p38MAPK, and NF-κB, and results in cancer progression and metastasis [146,147]. Recent studies have revealed that HMGB1-RAGE signaling triggers ERK-mediated mitochondrial Drp1 phosphorylation leading to autophagy for chemoresistance and regrowth in colorectal cancer [148]. RAGE also binds amyloid-β peptides (Aβ40 and Aβ42), which form aggregates in the brain and promote neurodegeneration [5] and may also act as mediators of disease progression in AD by interacting with Aβ and transporting it across the blood-brain barrier (BBB). In addition, these bindings result in the expressions of pro-inflammatory cytokines and endothelin-1 [66].

Quinolinic acid, a neuroactive metabolite of the kynurenine pathway, is an agonist of the N-methyl-D-aspartate (NMDA) receptor and is normally present at nanomolar concentrations in human brain and cerebrospinal fluid (CSF). Excessive quinolinic acid levels have been implicated in a variety of human neurological diseases, including AD and Huntington’s disease [67]. The direct interaction between quinolinic and the VC1 domain of RAGE is involved in early responses to noxious stimuli and may be associated with signaling cascades leading to cell death [6].

Endothelial RAGE interacts with Mac-1 on leukocytes [45], and Orlova et al. reported HMGB1 dose-dependently enhanced the interaction between Mac-1 and RAGE and induced the activation of NF-κB in neutrophils [47].

Lysophosphatidic acid (LPA) is a serum phospholipid with growth factor-like activities in many cell types. LPA stimulates cell migration, proliferation, and survival by acting on its cognate G-protein-coupled receptors [149]. Furthermore, RAGE was found to be required for LPA-mediated signal transduction, leading to proliferation and migration, in C6 glioma cells and smooth muscle cells [68].

Phosphatidylserine (PS) is a structural component of nuclear envelopes, endoplasmic reticulum, the inner cytosolic region of plasma membranes, myelin, and Golgi apparatus [150]. Cells undergoing apoptosis display PS on their surfaces, and this acts as a signal for the induction of phagocytosis and is recognized by receptors of lysosomal phagocytic vesicles. RAGE-deficient alveolar macrophages showed harmed apoptotic thymocytes and resulted in the defective clearance of apoptotic neutrophils in RAGE-deficient mice. PS-RAGE binding plays an important role in the GTPase and Rac1 signaling pathways. Details of the PS-RAGE interaction are still being explored and the molecular mechanism has not been determined [69].

Complement protein C1q binds to microbial surfaces or immune complexes, and thus, stimulates the complementation system and the productions of membrane lytic complexes, opsonins, and anaphylatoxins. Surface plasmon resonance (SPR) showed RAGE and C1q interact directly with a Kd of 5.6 μM, and this interaction plays a role in adaptive immunity and promotes C1q-mediated phagocytosis [70].

The recognition of DNA and RNA derived from hosts or pathogens is one way the innate immune system responds to infection and tissue damage. RAGE promotes DNA uptake by endosomes and augments TNR response through TLR9 [8]. TIRAP and myeloid differentiation primary response gene 88 (MyD88) are TLR2/4 adaptor proteins, and these proteins induce AKT, p38, IKKα, and JNK. Sakaguchi et al. demonstrated that bindings between phosphorylated RAGE and TIRAP or MyD88 result in the transduction of downstream signals [151]. Furthermore, RAGE-TLR crosstalk is involved in chronic inflammatory reactions and is considered a potential target for the treatment of neurodegenerative disorders [152].

### 3.2. Exogenous RAGE Ligands

RAGE plays a role in immune reactions by recognizing and responding to various PAMPs, including bacterial lipopolysaccharide (LPS), bacterial DNA, and viral and parasitic proteins. LPS is a major component of the cell walls of Gram-negative bacteria and initiates inflammatory cascades leading to sepsis. RAGE directly interacts with LPS and is involved in LPS-induced NF-κB activation and endothelial hyperpermeability [10,153]. The VC1 domain of RAGE can bind 22-nt dsDNA derived from vaccinia virus and a 22mer CpG motif-containing dsDNA [8]. Human respiratory syncytial virus (RSV) is a major cause of severe respiratory tract infections. The RSV fusion (F) protein mediates fusion between the viral envelope and airway epithelial cells. RAGE binds F protein and enables RSV infection of airway epithelial cells [11]. Longistatin secreted in saliva of the tick *Haemaphysalis longicornis* also binds to RAGE. Longistatin acts as a RAGE antagonist, suppresses NF-κB translocation, and thus, hosts immune response [12].

## 4. RAGE Ligand Signaling

Although many extracellular ligands interact with the VC1 domain of RAGE, the cytoplasmic domain of RAGE plays a vital role in RAGE-mediated signaling and overall RAGE function. RAGE activates diverse intracellular signaling pathways, including those of p38MAPK, AKT, ERK, mammalian diaphanous 1 (mDia1), and Rho GTPase (Rac1, Cdc42), and these pathways activate cascade transcription factors, such as NF-κB, SP-1, STAT3, and EGR-1 [7,51,154]. RAGE activates various signals by binding with adaptor proteins, such as mDia1, PKCζ, ERK1/2, dedicator of cytokinesis 7 (DOCK7), and DIAPH1 [13,14,15,16,151,155]. This interaction blockade presents a novel potential therapeutic target (Figure 2).

The cytoplasmic domain of RAGE binds with formin homology domain-1 (FH-1) of mDia1 [13], and the role of RAGE-mDia1 signaling is regulated during vascular smooth muscle cell (SMC) signal transduction and migration [156]. The S100B/RAGE/mDia1 pathway was reported to induce the migration of microglia via the activations of Rac1, JNK, and AP-1 and to result in the upregulations of the chemokines, CCL3, CCL5, and CXCL12 [157]. RAGE-mDia1 signaling also mediates the activations of Rac1 and Cdc42 during C6 glioma cell migration [15], and RAGE-mDia1 activates PKCβII, ERK1/2, and JNK signaling and regulates Egr-1 in hypoxic macrophages [158].

Ishihara et al. revealed by immunoprecipitation that ERK1/2 interacts with the cytoplasmic region of RAGE in HT1080 cells [16]. The release of HMGB1 from dying cancer cells enhances regrowth and chemoresistance via RAGE-ERK signaling, and the RAGE-ERK pathway activates the phosphorylation of Drp1 at residue S616, thus triggering autophagy for chemoresistance and regrowth in surviving colorectal cancer cells [148]. RAGE has four potential phosphorylation sites at Ser391, Ser399, SER400, and Thr401. Of these, only Ser391 is conserved in humans, mice, rats, guinea pigs, rabbits, cats, and dogs. When RAGE binds with one of multiple ligands, its cytoplasmic domain is phosphorylated by PKCζ. In HEK293 cells, RAGE phosphorylated at Ser391 co-precipitated with TIRAP or MYD88, and these interactions promoted downstream signal mediators, such as NF-κB, AKT, JNK, Rac1, and p38 [151]. Furthermore, in several types of cancer cells, DOCK7 (an XYZ) binds to the cytoplasmic domain of RAGE and leads to Cdc42 activation [155].

## 5. RAGE in Diseases

RAGE overexpression and activation are hallmarks of various diseases, including neurodegenerative, cardiovascular, vascular, and coronary diseases and atherosclerosis, diabetes, retinopathy, and cancer [58,159,160,161,162,163,164,165,166,167,168]. When a ligand binds to RAGE, downstream signaling pathways, including PI3K, ERK1/2, STAT, JAK, Rho GTPase, and transcription factors (AP-1 and NF-κB), are activated [15,19,166,169,170]. Furthermore, RAGE binding can also increase RAGE expression (Figure 3) [50].

### 5.1. Diabetes and Cardiovascular Disease

RAGE and its ligands accumulate in acute inflammatory conditions, such as diabetes, atherosclerosis, and nondiabetic vascular disease [58,159,161,171,172]. Diabetes mellitus is a disease of metabolic dysregulation resulting from defective insulin secretion, insulin resistance, or both. The RAGE-AGE pathway mediates vascular calcification and increases bone matrix protein levels through TGF-β, ERK1/2, fetuin-A, p38MAPK, PKC, and NF-κB [173]. AGEs significantly enhanced vascular intracellular calcium levels in rat bovine vascular smooth muscle cells (BVSMCs) [174], and induced cytosolic ROS production, which led to mitochondrial permeability transition and mitochondrial complex I deficiency in rodents [175]. Atherosclerosis is an inflammatory disease of the arterial walls, and RAGE has been linked to atherosclerosis development via several ligands, including AGEs, HMGB1, and S100 proteins [161]. Diabetes accelerates atherogenesis and RAGE deletion suppressed atherogenesis in ApoE null mice by activating the TGF-β/ROCK1 pathway [176]. Furthermore, AGE-RAGE accumulates in atherosclerotic lesions and increases the protein levels of MCP-1, PAI-1, VCAM-1, and ICAM-1 [177].

### 5.2. Neurodegeneration

RAGE has been reported to be elevated in human brain tissue in neurological disorders, including AD, Huntington’s disease, Parkinson’s disease, and schizophrenia [166,178,179,180,181]. RAGE-mediated transport of circulating Aβ across the BBB leads to Aβ accumulation and disruption of the brain’s vascular system [66]. HMGB1/HMGB1 receptors (TLR4 and RAGE) mediate the acute phase, during which damage to ischemic tissue and BBB permeability increase. In contrast, during the final phase of ischemic brain injury, HMGB1 promotes recovery and remodeling [182]. Immunohistochemical studies have demonstrated that RAGE levels are diminished in AD patients and that some of its ligands, such as AGEs, S100, and Aβ, lead to RAGE overexpression in neurons, microglia, astrocytes, and BBB vasculature [166,183,184,185,186]. Furthermore, in microglia, S100B/RAGE upregulated the Rac-1/JNK pathway and the transcriptional factors NF-κB and AP-1 [187].

### 5.3. Cancer

RAGE has been implicated in the pathogeneses of breast, bladder, hepatic, pancreatic, colorectal, gastric, and lung cancer, glioma, and melanoma [62,188,189,190,191,192]. RAGE is associated with various pathophysiological conditions and increased in cell migration and invasion resistance to apoptosis, autophagy stimulation, proliferation, and metastasis. Blocking RAGE signaling diminished tumor growth and proliferation in murine cancer models and offers an attractive means of targeting RAGE-mediated carcinogenesis [15,62,189,190]. AGE/RAGE pathways induced pro-tumorigenic proteins, such as ERK1/2 and cREB1 (cAMP response element-binding protein 1), and cancer progression and metastasis by MCF-7 breast cancer cells [193]. RAGE and its ligands also play vital roles in pancreatic ductal adenocarcinoma (PDAC) by increasing NF-κB activity and may be directly activated RAS which KRAS oncogenic mutations are observed in up to 30% of all cancers and in PDAC KRAS mutation is in nearly all tumors [194]. Interactions between RAGE and S100 proteins or HMGB1 are involved in melanoma progression and metastasis [195], and the AGE/RAGE pathway was found to increase the phosphorylation of ERK and promote tumor progression, invasion, and metastasis in gastric cancer via the RAGE/ERK/Sp1/MMP2 pathway [196]. HSP70 (heat shock protein 70) is actively released under inflammatory conditions and activates the inflammatory pathway. Somensi et al. demonstrated that HSP70 directly binds RAGE and stimulates ERK1/2, NF-κB, and TNF-α in human lung cancer cells A549 [197]. The AGE/RAGE pathway is also involved in carcinogenesis via RAS/ERK/Rac/CDC43 signaling [198]. PR3-RAGE binding mediates a signal transduction cascade involving the phosphorylations and activations of ERK/2 and JNK1 in prostate cancer cells [199]. Furthermore, elevated expressions of RAGE, thyroid transcription factor 1 (TTF-1), glucose transporter 1 (GLUT-1), and SOX2 were suggested to be early events during the development of HCV (hepatitis C virus) associated hepatocellular carcinoma (HCC) [200]. RAGE is also involved in the progression of pancreatic cancer in vitro and in vivo via the expressions of MMP2, MMP-9, NF-κB, and vascular endothelial growth factor (VEGF) [201], and in H1975 cells (a non-small cell lung cancer (NSCSLC) cell-line) was found to enhance growth, metastasis, and EMT (epithelial-mesenchymal transition) by activating the P13K/AKT and KRAS/RAF-1 pathways. Furthermore, in a H195 cell xenograft model. RAGE downregulation reduced tumor growth [202]. Thus, RAGE has been determined to be oncogenic and its involvement in diverse cancers has been well demonstrated, which suggests RAGE-ligand interactions offer promising therapeutic targets for RAGE-related diseases.

### 5.4. Other Diseases

Endothelial dysfunctions involve of the extracellular matrix (ECM) enzymes lysyl oxidase (LOX) and endothelin-1 (ET-1). At the gene level, the expressions of these enzymes are regulated by transcription factors such as NF-κB and AP-1. In human endothelial cells, AGE/RAGE increased the expressions of LOX and ET-1 through the AGE/RAGE/MAPK signaling cascade, which disrupted endothelial homeostasis by promoting cellular proliferation, altering the biomechanical properties of ECM, and impairing endothelial barrier functions [203]. In addition, uric acid (UA) induces endothelial dysfunction by inhibiting nitric oxide production. Cai et al. reported that human umbilical vein endothelial cells (HUVECs) exposed to high concentrations of UA overexpressed HMGB1, RAGE, NF-κB, and inflammatory cytokines. Furthermore, blocking RAGE significantly suppressed the upregulations of RAGE and HMGB1 [204].

RAGE and its ligands are also involved in coronary artery disease (CAD). sRAGE elevates acute ischemia and acts as a potential biomarker of acute coronary syndrome (ACS) [165]. Moreover, inhibition of RAGE using sRAGE protected against systolic overload-induced heart failure by modulating the AMPK/mTOR and NF-κB pathways [163]. 

Retinal microvascular dysfunction is a major component of diabetic retinopathy. RAGE plays a critical role in Müller glial activation and the downstream cytokine production associated with diabetic retinopathy [205]. In mouse models of type 1 and 2 diabetes, administration of sRAGE reduced early retinopathic abnormalities, such as endothelial and pericyte damage, loss of retinal neuronal function, retinal permeability, microgliosis, and inflammatory perturbation [206,207,208].

Obesity increases the risks of cardiovascular disease, hypertension, diabetes, and cancer. S100A4, S100A8/9, S100A12, and S100B act as DAMPs, activate receptors such as RAGE and TLR-4, and promote macrophage-based inflammation [209]. Furthermore, genetic deficiency of RAGE inhibited high-fat diet-induced weight gain, adipose tissue inflammation, energy expenditure, and insulin resistance [210].

Elevated RAGE expression in lung alveolar epithelial type 1 (AT1) cells may be involved in the proliferation and differentiation of pulmonary epithelial cells [211]. RAGE is expressed most in lung tissue and is an important mediator of diverse lung pathologies, such as pulmonary fibrosis, lung cancer, allergic airway inflammation (AAI), asthma, pneumonia, chronic obstructive pulmonary disease (COPD), bronchopulmonary dysplasia, and cystic fibrosis [168].

### 5.5. RAGE Polymorphisms and Inflammatory Disease

The gene coding for RAGE is located within the gene-dense major histocompatibility class III region on chromosome 6, which contains numerous genes involved in immune and inflammatory responses [212]. At least 30 polymorphisms within the exon, intron, and gene regulatory regions have been identified, and these polymorphisms affect RAGE expression and RAGE-mediated signals [213]. The major genetic variants of the RAGE gene involve a coding change in the V domain (Gly82Ser) and two changes in its promoter region (-429T/C and -374T/A) [41]. The gly82Ser isoform of RAGE exhibits enhanced ligand-binding affinity and increases inflammatory mediator levels. Gly82Ser genotypes are associated with elevated levels of serum AGE, serum CRP, plasma TNF-α, and urinary 8-iso-PGF_2__α_ [214]. The promoter region -439T/C variant of the RAGE gene acts as a biomarker of the diabetic/pre-diabetic state. In diabetic subjects, the -429T/C variant was associated with higher hemoglobin A_1c_ (HbA_1C_) levels. In addition, the -374T/A allele has been shown to affect gene transcription and to be a potential marker of vascular disease [215].

Although RAGE is greatly expressed in adult lung tissues, RAGE knockout (RAGE-KO) mice do not exhibit pulmonary changes associated with life expectancy [216]. However, RAGE-KO mice appear to have an extended life span, as more reached an age of 24 months than did wild-type mice in a comparative study [217].

sRAGE administration dose-dependently reduced aortic atherosclerotic lesion sizes, numbers, and complexities [41,218], and in sRAGE suppressed diabetic complications and inflammatory states [219]. Further studies are required to better understand the effects of RAGE isoforms and sRAGE.

## 6. RAGE Inhibitors

Interestingly, RAGE KO mice are healthy and developmentally normal, which suggests RAGE knockdown might be a safe therapeutic strategy [217]. Furthermore, extracellular ligand-based RAGE inhibitors have been shown to be effective in RAGE-mediated diseases. RAGE inhibitors are summarized in Table 2. TTP488 (azeliragon, also called PF-04494700; chemical name 3-[4-[2-butyl-1-[4-(4-chlorophenoxy) phenyl]imidazol-4-yl]phenoxy]-N,N-diethylpropan-1-amine) is an orally bioavailable small molecule that can cross the BBB [220]. TTP488 binds with multiple ligands, such as AGEs, HMGB1, CML, S100B, and Aβ [4,220,221]. TTP488 administration inhibited inflammatory signaling and neuronal Aβ accumulation in a mouse model of AD. While 10 weeks of treatment with TTP488 was found to be safe and well-tolerated in subjects with mild-to-moderate AD, TTP488 did not appear to show consistent effect on plasma levels of Aβ and inflammatory biomarkers [4,220,221]. Structurally, TTP488 presents two hydrophobic moieties, an aliphatic chain, and an electron-deficient aromatic group. TTP488 was developed by modifying the imidazole ring, the hydrophobic side group, and the aromatic core [222,223]. YS Lee et al. reported that a trisubstituted thiazole inhibited RAGE-Aβ interactions [222]. YT Han et al. discovered a novel series of 4,6-disubstituted 2-amino pyrimidines that act as RAGE inhibitors. SPR showed these inhibitors directly bind to RAGE and predicted the binding mode of 4,6-bis(4-chlorophenyl) pyrimidine analogs to the RAGE V-domain. Pyrimidine analogs also Aβ-induced NF-κB signaling in C6 glioma cells [223]. In later studies, a 4-fluorophenoxy analog improved RAGE inhibitory activity more than the parent 2-aminopyrimidine in vitro, and SPR confirmed direct binding between this analog and RAGE. Moreover, a 4-fluorophenoxy analog significantly reduced Aβ entry into the brain [224]. Furthermore, 6-phenoxy-2-phenylbenzoxazole derivatives that inhibit the RAGE- Aβ interaction in vivo and were not toxic to HT22 cells at 10 μM. These derivatives appeared to block Aβ transport across the BBB but did not seem to affect Aβ or amyloid plaque in the brain [225].

On the other hand, FPS-ZM1 (*N*-benzyl-*N*-cyclohexyl-4-chlorobenzamide) inhibits the interaction between the RAGE V domain and Aβ and the ability of Aβ to cross the BBB. FPS-ZM1 acts to block RAGE-mediated inflammatory signaling and inhibits nuclear NF-κB levels and BACE1, a key enzyme involved in the generation of Aβ in mouse brains. In addition, FPS-ZM1 had no toxic side effects in mice at doses up to 500 mg/kg [186]. Interestingly, treatment with FPS-ZM1 for 8 weeks attenuated cardiac remodeling and dysfunction in mice subjected to transverse aortic constriction (TAC), and treatment of TAC mice with FPS-ZM1 enhanced AMPK phosphorylation and reduced the phosphorylations of mTOR and NF-κB in cardiac tissues. In addition, treatment of TAC mice with FPS-ZM1 diminished endoplasmic reticulum stress, oxidative stress, and inflammation in cardiac tissues [163]. In a mouse model of toluene diisocyanate-induced asthma, FPS-ZM1 attenuated airway inflammation and β-catenin signaling [231], and in another study, FPS-ZM1 impaired primary tumor growth, prevented tumor angiogenesis and inflammatory cell recruitment, and most importantly, inhibited metastasis to the lungs and liver [190].

Chondroitin sulfate and heparan sulfate strongly bind to RAGE and suppressed the colonization of lungs by tumor cells [232], and GM-1111 inhibited interactions between RAGE and CML, HMGB1, and S100B and exhibited anti-inflammatory activity [226]. S100-derived peptide (ELKVLMEKEL) was found to compete for the RAGE site required for binding ligands, such as S100P, S100A4, and HMGB1, and reduced RAGE-mediated NF-κB activation, inflammation, tumor growth, and metastasis in different cancer cells [227]. In addition, peptides derived from the COOH-terminal motif of HMGB1 (150–183, 162–177, 160–183) also bind RAGE, inhibit the interaction between RAGE and HMGB1, and effectively suppressed the pulmonary metastasis and invasion of tumor cells [228].

Alagebrium (ALT7-11) is an AGE cross-link breaker, and treatment with alagebrium reduced AGE accumulation and atherosclerotic plaque formation and lesions [73]. AGE-RAGE signaling contributes to the development and progression of various diabetes and aging-related disorders. Yamagishi et al. found that DNA-aptamers might provide a potential treatment for vascular complications of diabetes and cancer by targeting the AGE-RAGE axis [229].

The intracellular domain of RAGE is required for many types of RAGE signals and for inducing downstream effects, and the disruption of RAGE-mDia1 binding is important for the therapeutic targeting of RAGE-mediated chronic inflammatory diseases. In one study, 13 compounds with high affinity for the cytoplasmic domain of RAGE (ctRAGE) inhibited the interaction between ctRAGE and mDia1 [14].

Recently found is that Src homology 2 domain-containing leukocyte protein of 76 kDa (SLP76) interacts with a cytosolic tail of RAGE, resulting in prompt activation of downstream signaling and gene transcription of pro-inflammatory mediators, such as TNF, CXCL10, HMGB1, and IL-6 in HEK 293 cells. Therapeutic targeting of interaction SLP76 with RAGE as a new approach for lethal sepsis [233].

Zheng et al. reported that aptamer-based antagonist against RAGE inhibits tumor growth and microvasculature formation in colorectal tumor mice by suppression of the RAGE/NF-κB/VEGF-A signaling [230].

## 7. Conclusions

RAGE is a multi-ligand receptor of the immunoglobulin family. RAGE and its ligands are present on most cell types and are involved in diabetes, diabetes complications, chronic inflammation, neurodegenerative disorders, and cancer. Recent advances have revealed the enormous breadth of the influence of RAGE and its ligands. Although, sRAGE and esRAGE act as biomarkers and endogenous protection factors against RAGE-mediated pathologies, sRAGE and esRAGE may not be ideal therapeutic means for targeting RAGE because they are large recombinant proteins that are difficult to produce at therapeutic levels. Accordingly, small-molecular inhibitors have been developed to target the extracellular ligand-binding site of RAGE and its intracellular signaling pathway. Further, there are critical issues that remain to be addressed of the understanding of RAGE-targeting therapy and the long-term impact of RAGE blockade in humans. Future investigations are required to improve understanding of the characteristics of efficient RAGE inhibitors to develop a significant understanding of the impact of RAGE blockage.

## Figures and Tables

**Figure 1 ijms-22-06904-f001:**
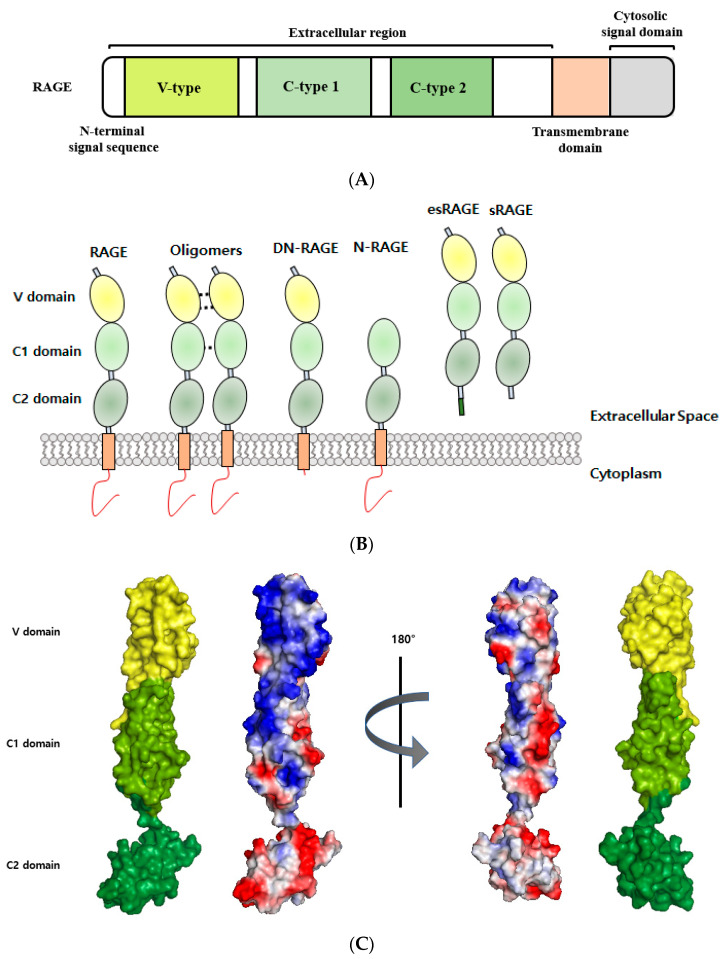
Structural analyses of RAGE. (**A**) Schematic representation of full-length RAGE domain. RAGE consists of a variable (V) domain, two constant (C1 and C2) domains, a transmembrane region, and a cytoplasmic tail. (**B**) RAGE isoforms. RAGE isoforms in the illustration include (from left to right) full-length RAGE, oligomers, dominant-negative RAGE (DN-RAGE), N-truncated RAGE (N-RAGE), endogenous secretory (esRAGE), and soluble form RAGE (sRAGE). (**C**) The surface of RAGE colored according to electrostatic charges (PDB ID: 4YBH). Positively charged areas are shown in blue, and negative charged areas in red. The figure was prepared using PyMOL.

**Figure 2 ijms-22-06904-f002:**
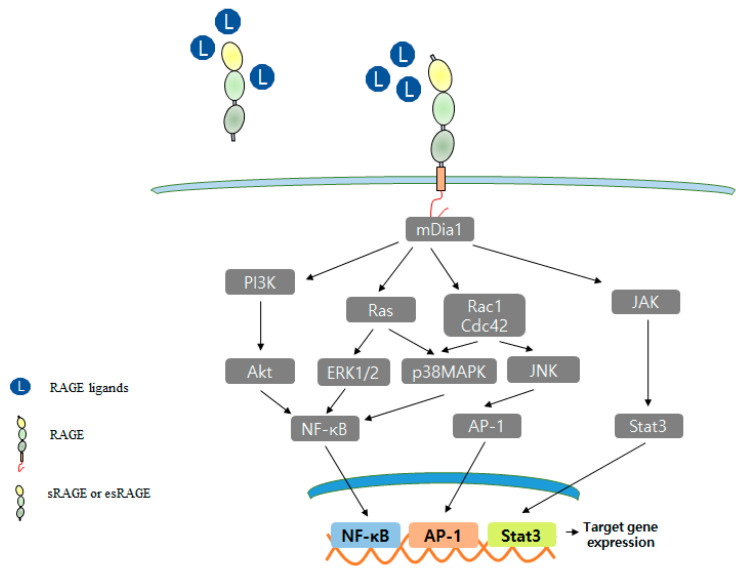
RAGE signal transduction pathway.
RAGE interacts with a diverse spectrum of extracellular ligands and multiple signal transduction pathways, including PI3K, p28MAPK, Rho GTPase, Rac1, and JAK pathways. At the transcriptional level, NF-κB, AP-1, and Stat3 have upregulated as vital targets of RAGE signaling, nevertheless other transcription factors.

**Figure 3 ijms-22-06904-f003:**
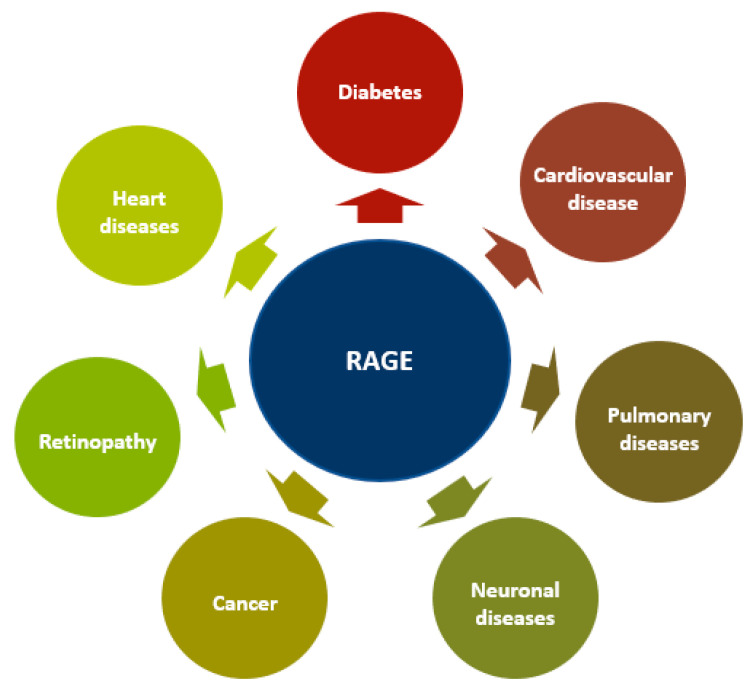
RAGE in different diseases. The overexpression and activation of RAGE are hallmarks of various diseases including neurodegenerative, cardiovascular, pulmonary and heart diseases, atherosclerosis, diabetes, retinopathy, and cancer.

**Table 1 ijms-22-06904-t001:** Significant RAGE ligands.

RAGE Ligands	RAGE Binding Domain	Clinical Significance	Ref.
Endogenous RAGE Ligands			
AGEs	V	Diabetes, chronic inflammation and cancer	[28]
S100/calgranulins	V or VC1 or V2	Inflammatory response and cancer differentiation and progression	[64]
HMGB1	VC1C2	Cancer development and metastasis and drug resistance	[65]
β-sheet fibrils	V	Neuronal disease: Alzheimer’s disease	[66]
Mac1		RAGE-mediated leukocyte recruitment	[45]
Quinolinic acids	VC1	Neuronal disease: Huntington’s disease	[67]
LPA	V	Cell proliferation and migration in C6 glioma and smooth muscle cells	[68]
PS		Rac1 activation in alveolar macrophages	[69]
C1q		Recruitment of leukocytes and phagocytosis	[70]
mDia1	cytoplasmic	Initiation and activation of RAGE-mediated signaling	[13]
Exogenous RAGE Ligands			
RNA or DNA	VC1	RAGE-mediated augmentation of inflammation	[8]
RSV F protein	VC1	Promote the survival of RSV-infected cells	[11]
Longistatin	V	Longistatin acts as an antagonist to RAGE and suppresses inflammation	[12]

AGEs: Advanced glycation end-products; HMGB1: High mobility group box-1 protein; LPA: Lysophosphatidic acid; PS: Phosphatidylserine; mDia1: Mammalian diaphanous 1; RSV: Respiratory syncytial virus.

**Table 2 ijms-22-06904-t002:** RAGE Inhibitors.

Inhibitors	Targeting of RAGE Domain	Effects	Ref.
TTP488	V	AGEs, HMGB1, CML, S100B, and Aβ-RAGE binding inhibition	[4,220,221]
4,6-disubstituted 2-amino pyrimidines	V	Aβ-RAGE binding inhibition	[223]
4-fluorophenoxy analog	V	Inhibition of amyloid plaques inside the brain	[224]
FPS-ZM1	V	Aβ-RAGE binding inhibition and low cytotoxicity in vitro and in vivo	[186]
GM-1111	VC1C2	CML, GMGB1, and S100B-RAGE binding inhibition	[226]
S100-derived peptide	VC1C2	Reduced RAGE-mediated activation of NF-κB, inflammation, tumor growth, and metastasis in various cancer cells	[227]
HMGB1-derived Peptide	VC1C2	Suppressed the formation of pulmonary metastasis and invasion in tumor cells	[228]
Alagebrium	AGE cross-link breaker	Reduced AGE accumulation and atherosclerotic plaque formation and lesions	[73]
DNA-aptamers		against the AGE-RAGE axis in diabetes-associated complications	[229]
Group of 13 compounds	cytoplasmic	Inhibition of ctRAGE interaction with mDia1	[14]
Aptamer-based antagonist	V	inhibit interaction between RAGE and S100B	[230]

AGEs: Advanced glycation end-products; HMGB1: High mobility group box-1 protein; CML: *N_ɛ_*-carboxymethyl-lysine; ctRAGE: cytoplasmic domain of RAGE.

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
