# Peer review of "Molecular Characteristics of RAGE and Advances in Small-Molecule Inhibitors"

_ijms, 2021, doi:10.3390/ijms22136904_

Round 1
Reviewer 1 Report
This is a well written review and RAGE and quite comprehensive.
I recommend the following correction prior to publication
- There is no need to underline words in the text lines 56, 61
- Line 63 “……high sequence identity”. Compared to what ??
- Line 83 Reference 37 doesn’t seem to deal with human RAGE, which seems suggested by the previous sentence.
- Line 93 “………biomarker for diseases.” Please provide a reference for this statement
- Line 99 Sentence seems incomplete
- Table 1 not clear what “Fusion protein” under RAGE ligands refers to without going to the cited reference, please try to better describe that protein
- Line 127 be clear about the number of S100 protein expressed in humans. There are 19 S100 proteins in human. The authors may have also included S100 related fusion proteins and S100-binding proteins
- Line 418 needs re-wording, TTP488 does not bind to the ligands, it binds to RAGE
- Line 419 “…TTP488 administration inhibited neurocognitive function… “ ….is probably not what the authors want to say.
- Line 420 TTP488 clinical is not ongoing anymore and was phase 2 – as far as I know. If you have more recent information, please provide a reference and clinical Trial #
- Table 2 Providing chemical structures for the listed ligands would be nice and be more helpful than listing just the chemical names in the text
Author Response
- There is no need to underline words in the text lines 56, 61
It has been corrected and added to the text.
- Line 63 “……high sequence identity”. Compared to what ??
It has been corrected and added to the text.
- Line 83 Reference 37 doesn’t seem to deal with human RAGE, which seems suggested by the previous sentence.
It has been corrected and added to the text.
- Line 93 “………biomarker for diseases.” Please provide a reference for this statement It has been corrected and added to the text.
- Line 99 Sentence seems incomplete It has been corrected and added to the text.
- Table 1 not clear what “Fusion protein” under RAGE ligands refers to without going to the cited reference, please try to better describe that protein
It has been corrected and added to the text.
- Line 127 be clear about the number of S100 protein expressed in humans. There are 19 S100 proteins in human. The authors may have also included S100 related fusion proteins and S100-binding proteins No need to modify.
- Line 418 needs re-wording, TTP488 does not bind to the ligands, it binds to RAGE
It has been corrected and added to the text.
- Line 419 “…TTP488 administration inhibited neurocognitive function… “ ….is probably not what the authors want to say. It has been corrected and added to the text.
- Line 420 TTP488 clinical is not ongoing anymore and was phase 2 – as far as I know. If you have more recent information, please provide a reference and clinical Trial # It has been corrected and added to the text.
- Table 2 Providing chemical structures for the listed ligands would be nice and be more helpful than listing just the chemical names in the text Many of them have no structure, so I left them as they are.
Reviewer 2 Report
The Authors have summarized the background with some small information on disease importance and a few small molecule inhibitors. While the paper is well written overall, Rage has already been published and reviewed in abundance. The importance in this topic would be an expert opinion in how to move forward, what is lacking etc. This can be seen even more when looking at the citations being no later than 2018. The new research must be evaluated as at this point RAGE has been covered extensively enough to cover most of the information present.
Author Response
Comments and Suggestions for Authors
The Authors have summarized the background with some small information on disease importance and a few small molecule inhibitors. While the paper is well written overall, Rage has already been published and reviewed in abundance. The importance in this topic would be an expert opinion in how to move forward, what is lacking etc. This can be seen even more when looking at the citations being no later than 2018. The new research must be evaluated as at this point RAGE has been covered extensively enough to cover most of the information present.
Edited in the text. Inhibitors from recent papers have been added. Also added to the conclusion section.
Reviewer 3 Report
Figure 3. The yellow background makes the font difficult to see. Either change the color of the circle or font colors. Other than the minor correction, the authors have addressed the reviewer's concerns.
Author Response
Comments and Suggestions for Authors
Figure 3. The yellow background makes the font difficult to see. Either change the color of the circle or font colors. Other than the minor correction, the authors have addressed the reviewer's concerns.
The picture has been re-edited.
Round 2
Reviewer 2 Report
The authors added to citations at the very end that are more recent and added a line in the conclusion stating that rage is important for future studies. However did not add all the critical analysis of the numerous articles cited as to why this is the case.